

# Metaverse as Concept

Jun Furukawa[1]

[1] Master's Student, Department of Engineering, Univercity of Tokyo

* jun.furukawa.baseball@gmail.com

## Abstract

In recent years, the term "metaverse" has been attracting a lot of attention due to Facebook's name change and other factors. However, this is not the first time that the metaverse has attracted attention; it had already gained momentum at the beginning of the 21st century. With this background, it can be said that the technology has already reached a certain degree of maturity and is entering the stage of social implementation. On the other hand, the social discussion on what it means for humanity and how society should deal with it remains immature. The purpose of this paper is to outline the metaverse as a concept to fill this gap. This paper will discuss how the Internet and VR have been positioned in society, while pointing out that they have closely overlapped but have maintained a delicate distance. It will then focus on cases and phenomena that cannot be distinguished between the two, and point out that the key to exploring the uniqueness of the metaverse lies in the subtle relationship between the two.

## Keywords

Metaverse, VR, Internet, Utopia

**Type**: Research article

**Citation**: F. Author et al. "How to write a peer-reviewed paper of the Journal of Architectural Informatics Society: ver. 20210329". Journal of Architectural Informatics Society, vol. 0, no. 0, pp. a1-aXX. doi: https://doi.org/xx.xxxx/xxxx/xxxxx

**Received**: 15 April 2020
**Revised**: 29 December 2021
**Accepted**: 05 January 2021
**Published**: 10 January 2021

## 1. Introduction

### 1.1. Background and objectives

Metaverse. In the past, it was merely the name of a virtual space service in science fiction novels and not a subject to be discussed on its own. However, with Facebook's name change to Meta Platforms in 2021, the term quickly became a household word and is now the focus of attention in many areas.

Metaverse is now a buzzword that has moved beyond being a virtual service in science fiction novels, and its exact outlines are difficult to grasp. Experts and organizations have begun to appear like bamboo shoots after the rain in the past couple of years, and they are engaged in heated debates over the definition of the term. Nevertheless, a common understanding has been established that metaverse and space are closely related. In addition, space is inseparable from the body that perceives it. Therefore, it is also commonly understood that the metaverse and the body are closely related. In fact, what is called a metaverse, whether Second Life, Fortnite, or VRChat, are a services based on a digital 3D space that is experienced through an avatar as a body. Space and body is one of the subjects of the architectural intellectual pursuits. Therefore, it was inevitable that Metaverse would seek architectural knowledge and that the architecture would be attracted to Metaverse.

In fact, star architects such as Kengo Kuma, Sousuke Fujimoto, and BIG have designed architectures in the physical world, and some architects such as Kanna Bansho and Fujito of xRaftnauts call themselves virtual architects or XR space designers. Furthermore, architectural competitions that uses virtual space as a site such as VRAA was held. As Kengo Kuma stated in an interview with Ken Sakamura[1], the demand for architectural designs that allow us to feel the body even in metaverse spaces will increase in the future.

The metaverse is in its infancy, and while the discussion surrounding the metaverse is immature, it is showing some degree of technological maturity.Therefore there is currently a large gap between the discussion and the progress of technological implementation. For example, problems such as avatars posted on VRoid, a platform for 3D models, being NFTed without permission, sexual harassment in the metaverse, etc. have occurred, but how to handle these issues is still unresolved in many areas. This is precisely the problem of "policy vacuum"

as pointed out by Moor [2]. In science technology in general, there has been a trend where hypothetical problems have been raised before some accident happen and discussed (e.g., the trolley problem with regard to automatic driving technology), but the current trend is toward instantaneous discussion of problems that have actually occurred. It can be said that the metaverse is on the verge of entering such a stage, and this is precisely why dense, deep, and long-lasting discussions are required.

Then what approach should architectural studies take to the metaverse? There are two issues that need to be addressed in the discussion of the metaverse in the field of architecture.

The first is that discussions are biased toward an engineering perspective. As evidenced by the recent establishment of the Architectural Informatics Society(AIS), various discussions are taking place in architecture with information technology in mind. For example, new design methods using information technology, and methods of describing and utilizing spaces using BIM and game engines. However, the metaverse is not only a place to be utilized for other purposes, but also an target to be entered and experienced, which is an objective in itself. In this respect, not only an engineering perspective but also a philosophical and social perspective is required.

Second, the discussions on real space that have been the subject of architectural studies up to now cannot be applied directly to digital space, thus the relationship between the digital and the physical is at the center of the discussion. Needless to say, such discussions are important, but if we are not careful, we run the risk of straying away from the essential issues. This is because the metaverse is not a digital 3D space or information space as a collection of data itself. Metaverse is a concept, and digital 3D space and information space are merely the representations it presents to us. Research on metaverse must first focus on metaverse as a concept, which is the premise for discussion. It is only with this premise that we can open the horizon for discussions of space, the body, and architecture in relation to the metaverse.

The purpose of this paper is to outline the contours of the metaverse as a concept, which is the premise of the discussion. Specifically, we will focus on the Internet and VR technology as representative technologies for constructing the metaverse, and examine the metaverse as a concept through a historical analysis of their technological development, the environment and ideas that supported it, and the debates and social movements related to it.

### 1.2. Structure

Chapter 2, as a preparation for the analysis in Chapter 3 and beyond, will refer to the discussion on utopia and indicate from what perspective we will discuss the metaverse in this paper. In Chapter 3, I summarize the circumstances surrounding the metaverse and point out that it is the successor of two trends: Internet technology and VR technology. Chapters 4 and 5 focus on the Internet and VR, respectively, and organize the historical context of how they have developed and been positioned in society. In Chapter 6 I organize the historical context of how Internet and VR technologies are related and how their convergence is recognized as a metaverse. In Chapter 7, I review the arguments of this paper as a future perspective, and point out the important issues in discussing the metaverse in the future.

## 2. Perspective of Utopia

### 2.1. Discussion over Utopia

In general, criticism has been leveled against technology optimists and idealists for being utopian. The main criticism is that they are simply escaping from reality and not facing the problems of the real world.

Although criticisms of utopia have been made from before, the situation regarding utopia has been further accelerated in the postmodern era. Zuk[3] and Araki[4] have argued that the end of the Cold War, the failure of the Eastern Bloc, and the failure of socialist attempts have spread the idea of "the end of history" and "the end of utopia". And within this context, in the context of conservative neoliberalism and techno-bureaucracy, anything that does not base its final value judgment on economic and technological efficiency is considered a utopia. Therefore, in modern times, people who supports the ruling oreder claims themselves as "realists" who come out against "utopians".

However, this interpretation of utopia is one-sided, and there is a movement to evaluate its function through analysis and examination of utopia. For example, Ito [5] acknowledges the value of utopianism, saying that when given a major goal, realism is effective in considering how to realize it with the current resources, but it is the role of utopianism to criticize reality and envision a better society. The target of criticism is the situation in which the conception of

such a utopia is becoming difficult. Wakabayashi[6], for example, expresses a sense of crisis that presentist progressivism today can only imagine the future as an extension of the present, and examines how it is possible to escape from the here and now.

As we have seen above, utopia has many aspects, and if not used carefully, it can confuse the discussion. Metaverse is often over-hyped as something that will create a new world and a new era. In this sense, it can be criticized as utopian. In fact, Daikoku[7] and Kizawa[8] have pointed out the utopian orientation of the Metaverse. However, in light of the above discussion of utopia, it is obviouse that utopia is a term that must be used with greater resolution rather than in a one-sided manner.

## 2.2 Utopia in this paper

This paper attempts to analyze the debate over technology in more detail by focusing on its functions. According to Levitas [9], utopia has three potential functions:compensation, critique and change. Utopia as a compensation corresponds to what Mannheim calls ideology. Mannheim [10] distinguished between ideology and utopia as consciousness that is incongruous with the state of reality, with ideology maintaining the status quo and utopia transforming it. It is difficult to distinguish between ideology and utopia, and it has been pointed out that it is often possible only after the fact.But it can be taken as suggesting the necessity and possibility of reevaluating various movements that were pointed out as utopian in the past through historical survey. Although it is difficult to distinguish between criteque and change, Levitas distinguishes between the two on the grounds that change requires the hope that it is feasible[9]. This position is the underlying tone that runs through this paper, and although we will not apply it to each case, I would like to emphasize that this perspective enables us to reevaluate what has been criticized as utopia, and to criticize what has been praised as realistic.

## 3. What is metaverse

In this paper, we treat metaverse as a concept. What do we mean by metaverse as a concept? It means that a metaverse is not a technology, a service, a digital space, or a community. Of course, there is no doubt that these elements are closely related to the metaverse, but each of these elements is not the metaverse itself. The metaverse is the whole of its parts. The metaverse may change with changes in technology, and the metaverse may change with changes in the community. And as the metaverse changes, services and digital spaces may change. Treating the metaverse as a technology, service, digital space, or community that conforms to a specific definition may mask the complex interrelationships that the metaverse as a whole contains. In order to grasp the complex situation surrounding the metaverse, it is essential to view the metaverse as a concept.

So what is the current state of the metaverse? The current state of the metaverse is chimeric. One of the reasons for the recent focus on the metaverse is Facebook's name change to Meta. Although the term "metaverse" had been used in some area before this, the massive attention it attracted in the business context forced the integration of previously disparate flows, and also invited the entry of flows that had been proceeding independently of the metaverse. The Metaverse is not a vague image formed by lines that progressed bottom-up, but a chimera that is the result of integrating everything. Therefore, the discussion tends to focus on each individual element, and the discussion of the metaverse as a whole has not progressed, which has caused confusion in the discussion. However, it is difficult to directly understand the metaverse as a concept, and it is impossible to discuss it without a rough definition of what it is. Therefore, let us first look at concrete examples of what has been called the "metaverse". Some concrete examples of what is currently called a metaverse include Second Life, Fortnite, and VRChat. Although there are minor differences such as first-person or third-person perspective, large scale or small scale, they all have one thing in common in that they share a space with multiple people online. In light of the above, the following definition by Yoichiro Miyake is appropriate.

Miyake [11] states that the broadest definition of a metaverse is "a digital space in which everyone can participate. In this paper, too, I adopt this broadest definition in order to grasp the breadth of the metaverse, on the condition that it is not the metaverse itself, but the representations that the metaverse presents to us. This definition can be broken down into two elements: "everyone can participate" and "a digital space for participation. The former part of the definition of the metaverse corresponds specifically to the Internet as a technology that connects people, objects, and information, while the latter part corresponds to VR technology as a technology for participation, e.g., immersion and expression in it. These two developments

have not developed completely separately; rather, they are closely related. Nevertheless, the distance between them is a delicate one. In this paper, we will first treat each as separate and organize their historical context. Then, the historical context will be reviewed, focusing on the relationship between the two.

## 4. History of the Internet

| 年代 | 出来事 |
|------|--------|
| 1957 | ARPA established |
| 1961 | Leonard Kleinrock comes up with the principle idea of packet exchange |
| 1962 | On-Line Man Computer Communicatoin(first Paper on the concept of the Internet) was published |
| 1964 | Paul Baran proposed a distributed scheme for U.S. telecommunications infrastructure with no central command or control point that would survive a "first strike" |
| 1969 | ARPANET is opened and four host computers are connected. |
| 1972 | The first pubic demonstration of network technology to the public |
| 1973 | Community Memory established |
| 1973 | Birth of Telnet, FTP, TALK |
| 1973 | Association for Computing Machinery(ACM) adopyed the code of ethics for their members |
| 1974 | TCP/IP invented |
| 1976 | Apple founded |
| 1976 | Walter Maner advocated the need for computer ethics |
| 1976 | Joseph Weizebnaum published "Computer power and guman reason" |
| 1977 | Emergence of early MUDs |
| 1978 | Launch of CBBS service, the first public dial-up BBS |
| 1979 | Usenet started |
| 1982 | Computer Professionals for Social Responsibility(CSPR) was established |
| 1983 | All networks connected to ARPANET use TCP/IP |
| 1984 | Junet operation started |
| 1984 | The first Macintosh is released |
| 1984 | ARPANET splits into MILNET and ARPANET |
| 1985 | Habitat, the original MMORPG (2D) appeared. |
| 1985 | "What is Computer Ethics" written by James Moor was published |
| 1985 | Computer Ethics Textbook was published by Deborah Johnson |
| 1989 | Birth of PSINet, the world's first commercial ISP |
| 1989 | WWW invented |
| 1989 | Robert Morris indicted for violating the Computer Fraud and Abuse Act (CFAA) (the first indictment under the Act) |
| 1992 | Birth of KEK, the first website in Japan |
| 1993 | WWW Specifications Published by CERN |
| 1993 | Birth of Mosaic, the world's first browser software created for browsing the World Wide Web |
| 1993 | HTML ver.1.0. released |
| 1994 | Brith of Yahoo! |
| 1994 | Amazon founded |
| 1995 | Window's 95 released |
| 1995 | Major carries such as British Telecom announced Internet services |
| 1995 | NSFnet was shut down completely and the American core Internet backbone was privatied |
| 1996 | "Hotmail" (browser-based e-mail service) launches in the U.S. |
| 1996 | "The Californian Ideology" written by Richard Barbrook & Andy Cameron was published |
| 1996 | Communications Decency Act passed in the U.S. |
| 1996 | The Declaration of Independence of Cyberspace |
| 1996 | Birth of google search |
| 1997 | Start of Ultima Online service |
| 1998 | Birth of Yahoo Bulletin Board |
| 1998 | microsoft begins antitrust litigation |
| 1998 | Google founded |

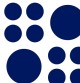

| 1999 | 2channnel launched |
|---|---|
| 2001 | The European Council finalizes its international Convention on Cybercrime and adopted it |
| 2001 | Wikipedia launched |
| 2003 | Myspace launched |
| 2004 | mixi launched |
| 2004 | Facebook launched |
| 2005 | Youtube launched |
| 2006 | Facebook opened to the public |
| 2006 | Twitter started |
| 2007 | Iphone released |
| 2007 | Start of Youtube partner program (initially vetted) |
| 2007 | Facebook introduces beacon feature |
| 2008 | Paper presented by Satoshi Nakamoto |
| 2010 | Complaints about the Right to be Forgotten |
| 2011 | Youtube partner program Opened to the public |
| 2016 | Trump won the election |
| 2018 | GDPR Enforcement |
| 2018 | Cambridge Analytica case |
| 2018 | "Algorithms of Oppression" written by Safiya Umoja Noble published |
| 2019 | "The Age of Surveillance Capitalism" written by shoshana Zuboff published |
| 2020 | California Consumer Privacy Act begins to apply |
| 2021 | Facebook Whistleblowing |

Figure1. History of the Internet

### 4.1. The beginning of the Internet

The Internet concept is said to have begun in a 1962 memorandum by J.C.R. Licklider. He envisioned a world in which computers around the world would be interconnected via the Net, allowing anyone to quickly connect to data and programs from anywhere [12]. His vision took a step toward realization in 1962 as an ARPA project, and in 1969 the ARPANET interconnected computers at four universities. In 1972, ARPANET was demonstrated for the first time to the public at the International Computer Communication Conference ( ICCC), and was internationally connected in 1973. The following year, in 1974, TCP/IP was born and connections between different networks became active. Then, in 1983, all networks connected to the ARPANET adopted TCP/IP, and since then, the set of publicly accessible, interconnected, TCP/IP-based networks has been called the Internet [13].

There is debate over two arguments behind the creation of the Internet. On the one hand, there is the opinion that it is a military situation [14][15], and on the other hand, there is the opinion that it has nothing to do with military circumstances [12][ 16]. What kind of otganization is ARPA, which led the research of th Internet? ARPA is the Advanced Research Project Agency, now renamed the Defense Advances Research Project Agency (DARPA). ARPA was launched as a response to Soviet's successful launch of the world's first satellite, Sputnik, in 1957. One position is that the development of Soviet technology increased the sense of danger of attack by long-range nuclear bombs, and the goal was to build a decentralized network to withstand such an attack [16]. The background for the spread of this theory is said to be the 1964 RAND study on robust decentralized networks [12]. However, this research took place six years after the launch of ARPA, and some argue that it cannot be said to have had a significant impact on the idea of ARPANET [12][16]. It is not clear what the actual idea was, and it may not have been entirely consistent with either side's position, but it is important to point out, nevertheless, that the Internet is still imagined today in relation to the military background.

### 4.2. The Internet in society

The Internet entered the commercialization phase in 1984 when ARPANET was split into MILNET for military and ARPANET for non-military use [16]. By 1985, the Internet was established as a technology that supported a community of researchers and developers and was beginning to be used by other communities as a means of everyday computer-mediated communication [12]. Internet technology had outgrown the laboratory and spread throughout society.

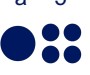

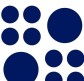

Attempts to communicate within a community via computer had been underway long before the Internet became widely available, and in the mid-1970s, the world's first computerized public electronic bulletin board, Community Memory, was installed in Berkeley, California. Community Memory was strongly influenced by the values of the California counterculture of the 1960s, the Appropriate Technology Movement, and the Silicon Valley hacker counterculture. They aimed to reduce the barriers between information technology and the general public, and operated by providing simple system training programs and placing terminals in public places such as libraries, although they could not be accessed over the Internet or modems. [17]. The formation of such Internet communities accelerated with the spread of the Internet.In 1979, Usenet, a system of computer mediated communication that introduced a bulletin board system, was developed[17][ 18], and in the mid-1980s, Listserv, a mailing list software, became popular[18][19]. For example, a face-to-face meeting was held in 1992 by the participants of the Social Work discussion list (SOCWORK), which is one of the listservs, which later developed into more serious academic conferences[14].At the end of the 1980s, while many people, mostly researchers, were using email, the Internet was not widely used by the public [16].

In 1989, the World Wide Web, a hypertext system offered on the Internet, was invented, and the world's first browser software for viewing it, Mosaic, was developed in 1993. In 1994, "Jerry and David's Guide to the World Wide Web," the predecessor of Yahoo! was started.In 1995, major carriers such as British Telecom announced Internet services.With this background, the Internet expanded explosively in the mid-1990s. By this time, connectivity was no longer an issue [16], and various developments were taking place. For example, the need to find information accurately and quickly led to the release of the first version of Google in 1996. In 1997, Ultima Online, the original 3D MMORPG, was released. 1998 saw the launch of Yahoo BBS, 1999 saw the launch of 2channel, and 2001 saw the launch of Wikipedia. In 2003, Myspace started its service, and by June 2006, there were more than 100 million Myspace users [16]. After that, mixi (2004), Youtube (2005), Facebook (2006), Twitter (2006), and other social networking services were born one after another, which are still very influential even today. SNSs then spread at an accelerated pace with the background of social events such as the Great East Japan Earthquake and the Arab Spring [20]. SNS has also been changing along with this trend, and there have been activities such as Youtubers who not only exchange information as in the past, but also earn income through such activities. In combination with these trends, the Internet has become a major concern for individuals, companies, countries, and all other entities.

### 4.3. Discussion on the Internet

In parallel with the development of the Internet, various debates and issues have been raised. Such discussions can be traced back as far as 1948, if we include computer technology. Norbert Wiener, one of the scientists active during World War II, had already warned about the negative aspects of technology in his book [22]. Through 1960s to 1970s, due to the increase in computer crimes and the threat of a surveillance state like Oceania in 1984, the Association for Computing Machinery adopted a code of ethics for their members, and laws concerning personal concessions and computer crimes began to be enacted mainly in Europe and the United States [22]. In 1976, Joseph Weizenbaum published "Computer Power and Human Reason," in which he discussed the differences between AI and humans and expressed a sense of urgency about entrusting important decisions to computers [23]. In the same year, Walter Maner proposed the need for Computer Ethics, and in 1985, a milestone essay, "What is Computer Ethics" by James Moor, was published. In his essay, he discussed why computers raise troubling ethical issues and proposed concepts such as policy vacuum and conceptual muddle [2]. 1985 was also the year that a textbook on computer ethics was published by Deborah Johnson [22]. Also in 1982, Computer Professionals for Social Responsibility (CSPR) was founded. They focused their discussions mainly on military use until the mid-1980s [16]. In the midst of this growing computer-related crisis, a computer worm developed by Robert Morris in 1988 shut down the nationwide computer network. As a result, he became the first applicant for the Computer Fraud and Abuse Act (CFAA) and was sentenced to a fine and probation [24]. Thus, it can be said that early discussions focused on ethical issues while at the same time turning their attention to the real issues of computer crime and military use. In the late 1990s, however, this trend began to change.

Such a change is typified by John Perry Barlow's A Declaration of the Independence of Cyberspace. This declaration was submitted as a statement of opinion in response to the 1996 passage of the Communications Decency Act. He strongly criticized that they are the inhabitants of cyberspace, that they have their own way of society and that the real world has no right to intervene in it [25]. On the other hand, there was also criticism of such cyber

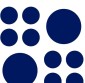

libertarian ideas. Richard Barbrook, for example, in a paper submitted in 1996 [26], described the Internet culture of that time as an amalgam of "the free-wheeling spitis of the hippies and the entrepreneurial zeal of the yuppies" calling Californian Ideology and criticized there for not addressing the real problems of racism, inequality, and environmental pollution that the United States was facing. In 2001, Amy Lynne Bomse, looking at the state of the debate, noted that many legal scholars consider the private, self-organizing regulatory nature of cyberspace to be a desirable alternative to the rule by law of the state. But on the other hand, she states that some strong criticisms have been made [27]. And it can be said that such debates are gradually moving toward a position that acknowledges the limitations of the Internet. For example, in "Internet democracy: Promises and limits," written by Dominique Cardon in 2010, the author sees in the web in the dusk created by the Internet the potential for a new kind of democracy, but also feels the danger that it will be neutralized by political and economic forces, and argues the need for a critical analysis of the Internet [28]. In the same year, "You are not a Gadget: A Manifesto," written by Jaron Lanier, refers to a group of people, mainly in Silicon Valley, who prioritize the Internet over humans and operate under a technological determinist ideology as "cybernetic totalitarians" and argued for the need for a more humanistic computer science [15]..
As described above, from the late 1990s to the 2000s, as the Internet became popularized, the debate shifted from topics such as crime and military use to more familiar issues of everyday life, such as politics, economics, and culture. This trend accelerated in the 2010s and beyond.

It could be said that technology companies such as GAFA are at the center of the current debate over the Internet. The social debate about giant technology companies had already been going on since the end of the 20th century, as seen in the microsoft antitrust law, but it has become more active since 2010. The most emblematic example of this is the "right to be forgotten" controversy , which began on March 5, 2010, when Mario Costeja González filed a lawsuit against the newspaper La Vanguardia, Google Spain, and its parent company Google Inc. over search results for his name on Google Search.As as result, the complaints against both Google companies were addmitted. The early 2010s, as mentioned above, was a time of solidified expectations for social networking, but the discussions over its negative aspects rapidly accelerated after that [20]. The use of social networking services for the 2016 U.S. presidential election, the related Cambridge Analytica case, and Facebook in 2021, to name but a few, generated much controversy. In parallel, there have been many controversies, with GAFA in mind, such as the "algorithms of oppression"[29] which criticized the google search engine for its algorithm that encourages discrimination, and "The Age of Surveillance Capitalism" [30], which warned that personal information is being exploited like a natural resource, and that our future actions are being prompted and commodified by them.The debate on internets since 2010 is focusing on the activities of global corporations, which have been growing in size, involving all individuals, corporations, and nations.

## 5. History of VR

### 5.1. The Beginnings of VR

The world's first VR system is said to be Sensorama, developed by photographer and film expert Morton Heilig in 1961 [31]. The Sensorama was an enclosure system that consisted of a full-color stereo image, a fan, a scent spur, a stereo sound system, and a moveable chair that allowed visitors to experience things like riding a motorcycle through New York City or riding in a helicopter. It is interesting to note that its birth was very different compared to the Internet. However, VR systems, like the Internet, would go on to make great strides in the laboratory. Ivan Sutherland was an indispensable figure here. In 1963, he developed the "Sketchpad," a system for drawing on computer screens using a light pen, and in 1965, a paper entitled "The Ultimate Display" claimed that a computer-connected display was a mirror of a mathematical wonderland that would never be realized in the physical world, and envisioned a form of VR that anticipated today's VR [32]. And so was born in 1968 the first Head Mounted Display, The Sword of damcles. It was much larger than today's HMDs, but the head-mounted version was revolutionary.

However, various forms of VR systems were developed in the following years. In 1977, Tom Furness introduced the idea of the "virtual cockpit. In 1977, Tom Furness introduced the idea of a "virtual cockpit," a virtual three-dimensional panoramic "bubble" that displayed all the information the pilot needed. The first system, called the "Visually Coupled Airborne Systems Simulator" or VCASS, was implemented in 1982 [33]. In 1978, the Architectural Machine Group led by Nicholas Negroponte developed a system called the Aspen Movie Map. The Aspen Movie Map was a video recording of the town of Aspen, and by touching the screen to indicate the direction you wanted to go while viewing the town on the TV screen, the scenery

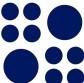

in that direction would be projected. At the time, terms such as artificial reality, telexistence, telepresence, and virtual environment were used in different fields. However, at the Santa Barbara Conference held in 1990, virtual reality was adopted as the generic name for all of them [34].

| 年代 | 出来事 |
|------|--------|
| 1961 | The first VR system, Sensorama, was developed by Morton Heilig |
| 1963 | "Sketchpad" was developed by Ivan Sutherland |
| 1965 | "The Ultimate Display" was presented by Ivan Sutherland |
| 1968 | The first HMD, The Sword of damocles, was developed by Ivan Sutherland |
| 1968 | "Do Androids Fream of Electric Sheep?" written by Phikip K Dick was published |
| 1977 | Tom Furness introduced "Virtual Cockpit" |
| 1978 | "Aspen Movie Map" was developed by Architecture Machine Group |
| 1982 | "Buring Chrome" written by William Gibson was publidhed |
| 1982 | "Virtual Cockpit" was implemented |
| 1984 | "Neuromancer" written by William Gibson was published |
| 1984 | Jaron Lanier established VPR research |
| 1989 | RB2, the world's first commercial VR, was announced; the HMDwas called eyephone |
| 1990 | Conference of Santa Barbara was held |
| 1992 | "Snow Crash" written by Neal Stephenson was published |
| 1994 | Paul Milgram's proposal of the "MR" concept |
| 1995 | Nintendo announces Virtual Boy |
| 1996 | Sony announces Glasstron |
| 1996 | The Virtual Reality Society of Japan was established |
| 1996 | Debut of Kyouko Date |
| 1999 | "The Matrix" directed by the Wachowskis released |
| 2003 | Second Life lauched |
| 2004 | Unity established |
| 2004 | Roblox invented |
| 2007 | Birth of virtual singer Hatsune Miku |
| 2007 | google street view launched in the USA |
| 2007 | Second Life boom in Japan |
| 2007 | "Den-noh Coil" directed by Mitsuo Iso was aired |
| 2008 | Metaverse Associarion established |
| 2009 | "Summer Wars" directed by Mamoru Hosoda was released |
| 2009 | "Sword Art Online" written by Reki Kawahara was published |
| 2012 | Oculus founded |
| 2014 | Facebook acquires Oculus |
| 2014 | Alpha launch of VRChat |
| 2014 | Closing of Metaverse Association |
| 2016 | The Frist Year of VR |
| 2016 | Pokemon GO launched |
| 2016 | Birth of Kizuna Ai |
| 2017 | Nijisannji, Hololive sestablished |
| 2017 | Fortnite launshed |
| 2018 | The 1st Vket held |
| 2018 | "Ready Player One" directed by Steven Spilberg was released |
| 2018 | VRM was invented |
| 2019 | DJ Marshmello held music concert in Fortnite |
| 2021 | Facebook name change to Meta Platforms |
| 2021 | NFT for digital artist BEEPLE's work "Everydays-The First 5000 Days" sold for $7.5 billion. |

Figure 2. History of VR and metaverse

## 5.2. VR in Society

VR technology left the laboratory in 1989, just as the World Wide Web was invented. In that year, the world's first commercial VR system, the RB2, was launched by Jaron Lanier, which allowed two people wearing an eyephone to talk to each other in a virtual space. Lanier states that his purpose in creating RB2 was to enhance the imagination and expressiveness of the world, to make a world that people can relate to, and that people find interesting [15]. What is interesting here is that he does not think of VR as a reproduction of reality, but rather as linked to expression. In "The Idea of the Virtual," written in 1993, Quéau also asserted that the virtual is écriture, and it can be seen that he did not consider VR as something that one-sidedly makes the user feel a sense of reality, but rather as a means of expression [33]. This way of thinking seems to be still influential today, while maintaining a delicate distance from academic VR research.

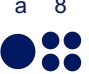

After the release of the eyephone in 1989, new HMDs were introduced: the Virtual Boy by Nintendo in 1995 and the Glasstron by Sony in 1996. However, they were not successful due to lack of technology, etc. HMDs became popular in 2016, which is also called the first year of VR because various HMDs including the Oculus Rift, HTC VIVE, and Playstation VR were released at a lower price than before.

The year 2016 is also important in a cultural sense, in the sense that the first virtual Youtuber, Kizuna Ai, began her activities. In Japan, 2007 saw the birth of Hatsune Miku, a virtual singer who is still known around the world today. Further back, in 1996, a virtual idol named Kyoko Date, who was created using 3D computer graphics and was treated as a talent affiliated with Hori-pro, made her debut. In this sense, it can be said that virtual personalities were active in Japan even before the spread of VR. Nevertheless, it is unprecedented in that Vtubera, an entities strongly associated with the word "virtual," is now operating beyond the single platform of Youtube, and is distinctly different from existing virtual entities.In 2017, NijiSanji and Hololive, which is Vtuber group, were born, and to this day the number of Vtubers are continues to grow. The fact that 28 out of the top 30 Youtubers in the world in terms of Youtuber superchat amount are Vtubers shows the extent of their influence [35].

### 5.3. Discussion on VR

We have looked at the technological development of VR and its relationship to culture above. Ethical and social discussions on the subject of virtuality are few. This is partly due to research difficulties in making a clear distinction between the Internet and VR, but it also seems to be related to the fact that VR is less widespread in society than the Internet. Even in such a situation, the issues raised by Quéau are important: In 1993, Quéau discusses the impact of virtual on society and points out its potential. At the same time, Quéau points out the danger of mistaking virtuality for reality, of considering reality as an extension of virtuality, and of mistaking virtuality as a referent and the possibility of a similar approach in reality [33]. Although the discussion on virtual reality is far from mature, it raises important issues that are pertinent to the discussion of the metaverse.

## 6. History of metaverse

### 6.1. VR and the Internet

As has been noted above, the Internet and VR are closely linked in many respects. For example, Ivan Sutherland, who laid the groundwork for VR research, has been heavily involved in Internet technology research; Licklider appointed Sutherland as his successor for Internet research at ARPA [12], and when the ARPANET connected four universities in 1969, Sutherland was working with Robert Taylor on methods of 3-D representations over the net[12]. Jaron Lanier, the father of VR, also recalls the time of the development of the first commercial VR, announced in 1989, as the world's first attempt to connect people in the world of virtual reality[15]. Thus, VR and the Internet are technologies that have developed without a clear distinction. The metaverse is a concept that has been fostered and conceived in the delicate relationship between the two. In order to analyze this process, it would be effective to look at the connection with the cultural excitement of science fiction works and services. But first, let us take a look at the social position of the metaverse.

### 6.2. Two Metaverse Booms

As mentioned above, the term "Metaverse" spread rapidly following Meta's name change in 2021. This is the second time, however, that the Metaverse has attracted attention.

According to Miyake[11], metaverse creation began around the year 2000, when the Internet was introduced to society, followed by Second Life in 2003 and roblox in 2006. The first time the metaverse attracted attention in Japan was when Second Life became popular in 2007 [36]. In response, the Metaverse Association was established in Japan in 2008. In the latter half of 2000, movies and novels with various metaverse motifs appeared, as described below. However, the fever of the Metaverse cooled down, partly because the technology was still in its infancy[36]. The closing of the Metaverse Association in 2014 is a true indication of this.

Although the fever has cooled, the concept of a metaverse continues to develop at the grassroots level, with the launch of a variety of services in the mid-2010s that are now attracting attention as metaverses, and social interaction with these services becoming active in the latter half of the 2010s. Then, Covid-19 brought VR into the spotlight, and various attempts were

made, such as the Virtual Tour de France and Virtual Shibuya.

In terms of the level of attention in society, metaverse once enjoyed a major surge in the mid-2000s, and then slowed down in the first half of the 2010s. Then, in 2021, it rose again and has continued to the present day.

### 6.3. Metaverse and Culture

As mentioned above, services called metaverses did not begin to be developed until the beginning of the 21st century, but the term metaverse dates back to 1992. It was the name of a fictional virtual space service in the science fiction novel Snow Crash by Niel Stevenson. Snow Crash is a representative of the science fiction genre known as cyberpunk [37].
Cyberpunk is a science fiction genre set in cyber space that was mass-produced in the 1980s and 1990s, starting with William Gibson's Neuromancer, published in 1984[37][38]. It is characterized, for example, by its wild science fiction imagination and its outlaw technology as its main characters, such as hackers and other criminals in the information society[38]. It has also been noted that it is often regarded as a dystopian genre [39]. The cyberpunk theme of a devastated near future is not limited to novels, but is also common in Blade Runner, The Matrix, and other works.

On the other hand, Japanese video works with metaverse motifs seen since the late 2000s have a very different style. For example, the anime "Den-noh Coil" by Mitsuo Iso, released in 2007, depicts an ordinary world, except for the fact that the world of the cybernetic world can be seen through cybernetic glasses. The main characters are children, and while the story sometimes depicts the runaway behavior of the cybernetic creatures, it also depicts their friendship. 2009's "Summer Wars" a film by Mamoru Hosoda, like Den-noh Coil, depicts an ordinary world except for the existence of "OZ", the virtual space on the Internet. The main character is a timid high school boy in love. In the latter half of the story, the world is on the verge of an unprecedented crisis caused by a runaway AI, but the world is saved by connecting people around the world via OZ.

In parallel with the birth of various novels and video works related to the metaverse, various services have also been created.The first known metaverse was probably Second Life, which began service in 2003. Second Life is a 3D digital space service in which players from around the world participate via avatars; items and land are bought and sold in Second Life, which has earned some people the equivalent of US$100,000 in their first year in Second Life[40]. There are also online newspapers, like The Alphaville Herald, the Metaverse Messenger and Second Life Newspaper, that deal with life in virtual life, accessible both on the Internet and in Second Life[41].In 2007, Second Life boomed in Japan and, as is the case with the Metaverse today, a large number of introductory books on Second Life and how to make money in Second Life were published.

As mentioned above, the metaverse lost its prominence, but even before it regained attention in 2021, various services with metaverse-like concepts, such as VRChat, PokemonGo, and Fortnite, were born, not necessarily tied to the metaverse, but have attracted attention. In addition to the launch of services, recent years have also been characterized by the various events held on the platform, such as Vket held on VRchat in 2018 and the music concert by DJ Marshmello held on Fortnite in 2019. Another difference from the first boom is the presence of blockchain technology. Whether or not blockchain technology is considered an essential part of the metaverse is a matter of opinion [36][42][43], but it is an element that cannot be ignored in that it is discussed in connection with the decentralized Internet. It is not clear at this stage how this will affect the concept of a metaverse, but the idea beyond its technical characteristics will need to be watched closely.

As we have seen above, the metaverse, while showing its own development, is closely related to the development of the Internet and VR. In light of this, it would be useful to rethink the metaverse with reference to the discussions that have taken place in the development of the Internet and VR. For example, with Quéau 's criticism of the inversion of reality and virtuality [33] in mind, it is obvious that the idea of conducting social experiments or experiments that would be costly in reality in the metaverse, or the idea that the real world mimics the metaverse [35][44][45] must be reconsidered.Discussion on Metaverse is biased toward an engineering, business, or policy perspective, there are also active cultural and social movements, and by paying attention to them, we may be able to find important issues to discuss.

## 7. Future Prospect

While asserting the importance of viewing the metaverse as a concept, this paper analyzes the development of the Internet and VR and then analyzes the development of the metaverse. Since

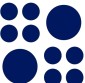

the main focus of this paper was on the historical development of the metaverse, it was not possible to analyze in detail the state of the metaverse today. In the future, it will be necessary to analyze not only technical discussions, but also how they are discussed from the perspective of business and culture, and what are the ideas behind them, using so-called business books and blogs as primary sources. It will also be necessary to consider how the ideas surrounding such a metaverse are represented in space by mobilizing the knowledge of architecture and sociology.

## Declaration of competing interests

The author declare no potential conflicts of interest with respect to the research, authorship, and/or publication of this paper here.

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

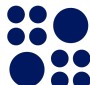

抄録

近年 Facebook の社名変更などの影響で「メタバース」という言葉が注目を集めている。だがメタバースが注目を浴びるのは今回が初めてではなく 21 世紀の初めには一度盛り上がりを見せていた。そのような背景もあり既に技術的にはある程度の成熟を見せており、社会実装の段階に入りつつあると言える。一方でそれは人類にとって何を意味するのか、社会はどのように向き合っていけばいいのかなど、社会的な議論は未成熟のままである。このギャップを埋めるために概念としてのメタバースの輪郭を描くのが本論文の目的である。本論文ではインターネットと VR が社会の中でどう位置付けられてきたかを論じつつ、それらが密接に重なり合いながらも微妙な距離感を保ち続けてきたことを指摘する。その上で、その両者のどちらとも判別のつかない事例や現象に注目し、その微妙な関係性にメタバースの独自性を探る鍵があることを指摘する。

