# OpenReview forum: "Metaverse as Concept"
_AIS-J.org/2021/Journal_

### Official Review · AnonReviewer2 · 2022-12-05

**Rating:** 3
**Confidence:** 4

**Review:**

There is no doubt that the metaverse, which this paper aims to deal with, has been rapidly gaining social attention in recent years, but on the other hand, considering that academic discussions on its essence and problems are still in their infancy, research and discussion on issues such as the one the author aims to address are necessary, so it is fair to say that the novelty of the topic is there. However, the results of research that goes beyond raising such issues are noteworthy.
However, the limitation of this paper is that there are unfortunately few research results beyond the raising of such issues.
The discussion of utopia in chapter 2 does not clearly state how it relates to the concept of metaverse. chapter 3 raises the issue of the absence of cultural context in the metaverse discussion, which is understandable in itself, but the significance of the discussion should be evaluated depending on its landing point. chapter 4, the history of the Internet, and chapter 5, the history of VR, are very interesting in their own right. Chapter 5, the history of VR, summarizes the main points very carefully, but it also lacks a discussion of the current metaverse from the perspective of the current metaverse. And even in chapter 6, the relationship with this historical analysis is not clear. As a result, it is difficult to say that there is sufficient significance to outline the contours of the metaverse as a concept, which is the premise of the discussion.
In the end, in light of this historical background, this paper argues that Discussion on Metaverse is biased toward an engineering, business, or policy perspective, there are also active cultural and social movements, and by paying attention to them, we can see that the Metaverse is a concept that is not only a concept, but also a premise for the discussion. In the end, given this historical background, what this paper argues is that Discussion on Metaverse is biased toward an engineering, business, or policy perspective, and by paying attention to them, we may be able to find important issues to discuss. However, if we are to consider metaverse as a concept from an ideological and cultural perspective, in addition to the issues discussed here, it seems more essential to discuss the appeal of metaverse beyond simple remote communication as an extension of human perceptual capabilities or as a substitute for real space, and the ethical issues it faces in relation to the existing society. It seems more essential to the discussion to take into account the ethical issues it faces with the existing society. For example, the dependence of human self-identity formed through social relationships on experiences in the metaverse, and the changing concept of spatial cognition itself, which should have relied on bodily sensations and their limitations and characteristics.
Thus, while we appreciate the topic that this paper addresses as a problem statement and the efforts to approach it in itself, we believe that at this point it is too early to say that the results obtained from it are significant enough to merit adoption of the paper. We look forward to further development of this research in the future.

**Overview:**

The theme of the paper and its significance in building informatics is understandable, but the paper does not reach the intended outcome.

---

### Official Review · AnonReviewer1 · 2022-12-06

**Rating:** 4
**Confidence:** 4

**Review:**

Metaverse is a concept, and digital 3D space and information space are merely its representations. Based on this perspective, this paper states that it is essential to consider an ideological and cultural perspective to discuss the metaverse, not just from the current engineering or business. While this assertion is understandable, the specific ideological and cultural perspectives from which to discuss the metaverse are not discussed in this paper. In other words, the paper merely points to what is lacking from the ideological and cultural perspectives. For example, it needs to discuss how the definition of utopia relates to the metaverse. Chapters 4 and 5 look back at the history of the Internet and VR, but again, only the problem is addressed. The positioning of the appearance of these histories in this paper needs to be more precise; otherwise, it could be confusing to the readers as to why these descriptions are in place. Chapter 6 likewise briefly reviews the history of the metaverse, but again, it needs to be clarified how this historical review relates to the central theme of this paper. From what specific ideological and cultural perspective should the metaverse be discussed, and why? The paper's significance can be found only when the claims are supported by evidence. We look forward to further development.

**Overview:**

The paper does not provide convincing reasons for claims or evidence to support its claims.